# Exploring if Gamification Experiences Make an Impact on Pre-Service Teachers' Perceptions of Future Gamification Use: A Case Report

Laura Guerrero Puerta [1,2]

1 Departamento de Didáctica y Organización Escolar y DDEE, Universidad Nacional de Educación a Distancia, 28015 Madrid, Spain; laura.guerrero.puerta@edu.uned.es
2 HUM-308 Research Group, Universidad de Granada, 18071 Granada, Spain

**Abstract:** Gamification has become an increasingly used pedagogical approach in the classroom, motivating students and enhancing their educational experience. This has led to the need for specific training for teachers, as well as a need to understand how this training can be effective and how contact with gamification during teacher training can influence the attitude of pre-service teachers. Therefore, this case report has focused on examining pre-service teachers' perceptions of gamification as a tool used in their teacher education. Qualitative semi-structured interviews were conducted to collect data on the participants' individual experience with gamification and their perception of future uses of this technique. The data were collected between December 2010 and February 2020 in Granada, Spain, and were analyzed using qualitative content analysis to identify general trends and findings. Pre-service teachers present a positive perception of gamification as a tool to enhance and motivate learning, highlighting the importance of prior training for its application in the right context, as well as the need to investigate other teaching approaches to improve the effectiveness of gamification. In conclusion, it should be noted that gamification is a pedagogical approach increasingly used in the classroom, so specific training is needed for the teachers responsible for carrying out this technique. This training should include research on the effectiveness of gamification, as well as the best approaches for its application, for student motivation, and for the creation of equal relationships between students.

**Keywords:** gamification; teacher training; higher education; motivation; teaching innovation





## 1. Introduction

In recent times, gamification has become an increasingly used pedagogical approach for teaching in the classroom. (Gamification, as articulated by Werbach and Hunter [1], is a strategic approach that involves incorporating elements and design techniques commonly found in games into contexts that typically are not game-related. This concept draws inspiration from the achievements of the gaming industry, the dynamics of social media, and extensive research in human psychology. Essentially, gamification can be applied to any task, process, or theoretical scenario to enhance engagement and motivation. Its primary goal is to encourage active participation by integrating game elements like rewards, awards, and leaderboards. To gain a deeper grasp of gamification, it is essential to dissect its key components, as highlighted by Sailer, Hense, Mandl, and Klevers [2]. These components include "game", "element", "design", and "non-game contexts". "Game" encompasses various situational elements, such as goals, rules, feedback mechanisms, and voluntary participation. "Element" distinguishes gamification from serious games, focusing on the specific components employed. "Design" centers on the use of game design principles rather than broader game-based technologies. "Non-game contexts" denote the wide range of applications beyond traditional gaming scenarios). This approach is based on the use of game elements to motivate students and enhance their educational experience [3,4]. This

trend has led to the need for specific training for the teachers in charge of carrying out this approach [5,6]. This makes it necessary to understand how this training can be effective and how contact with gamification during teacher training can influence the attitude of pre-service teachers in their future [7,8]. Therefore, this case report has focused on examining pre-service teachers' perceptions of gamification as a tool used in their teacher training. In particular, the study aims to understand how this training can be effective and how exposure to gamification during teacher training can influence the attitudes of pre-service teachers towards it in their future careers.

This case study design explores the influence of contact/experience with gamification on prospective teachers' perceptions of future uses of this approach. It was conducted at the University of Granada (Spain), in the master's degree program in Teacher Training in Secondary Education, Vocational Training and Baccalaureate. An attempt was made to identify students who had experienced gamification during their teacher training process, but this was difficult. Only three students were found who met these conditions and who wanted to participate in the study. Given the impossibility of continuing the study for reasons beyond the researchers' control and having been able to access a limited number of participants, it is presented as a case report, which presents mid-range theories, which, although they contribute to the gamification literature, should not be generalized, but rather taken as hypotheses for future studies. This decision has been made, as some of the results provided by the analysis of these three individuals are novel and deserve the opportunity to be further explored.

It is important to note that this study aligns with the established legal framework of the EHEA (European Higher Education Area), which places a strong emphasis on key principles such as educational excellence, student-centered learning, employability, and the acquisition of essential competencies. By delving into the potential of gamification within the context of teacher education, this research significantly contributes to the overarching goals of the EHEA. It provides valuable insights into innovative approaches that can potentially elevate the effectiveness of teacher training programs throughout Europe.

The EHEA serves as a comprehensive framework and a set of guiding principles for higher education across Europe, with a primary focus on nurturing educational excellence, facilitating student mobility, bolstering employability, and fostering the development of crucial competencies tailored to the demands of the 21st century [9,10].

## 2. Theoretical Framework

Gamification is a pedagogical approach based on the environment provided by video games, which seeks to increase the motivation levels of the users who participate in it, while improving the productivity levels of the company or institution. This pedagogical approach is based on the classic characteristics of games, such as their self-motivating and regulatory nature, adding to them a series of activities and objectives that promote the acquisition of skills through problem solving [3,4,11,12].

The literature extensively discusses the significance of incorporating games in the field of education, particularly within the framework of game-based learning (GBL) [13–19]. GBL is recognized for its potential to enhance teaching practices by utilizing games as a motivating and engaging instructional tool. Games have been found to naturally motivate students, providing a fun and interactive approach to learning compared to traditional methods perceived as monotonous [20]. This observation aligns with various psychological theories of learning, including sociocultural perspectives, which suggest that games can expand the zones of proximal development at the socioemotional and cognitive levels by stimulating users to perform beyond their cognitive capacities and engage in logical thinking in everyday situations. According to Piagetian constructivism, games not only reflect existing cognitive structures but also facilitate new learning, making them highly relevant to children's developmental processes. The contemporary literature increasingly explores games set in virtual environments or those based on them, and gamification,

characterized by the integration of tokens, challenges, and visual settings, falls within this category [3,4].

Moreover, recent studies have focused on how today's youth seem to significantly benefit in their learning processes when these games are based on transmedia narratives. Transmedia narratives refer to a narrative approach in which a story or narrative universe extends and develops across multiple platforms and media. Instead of being limited to a single medium, such as a book, a film, or a video game, transmedia narratives utilize diverse media, such as books, films, television shows, websites, social media, and games, to tell a complex and enriching story. Each medium contributes to the narrative in a unique and complementary way, expanding and enhancing the viewer or reader's experience. Transmedia narratives encourage active audience participation, inviting recipients to explore, interact, and contribute to the narrative world in various ways, providing a more immersive and participatory experience [21].

In his work, Jenkins [21] introduces a set of seven principles that provide a framework for understanding transmedia narratives and are closely related to the pedagogical power that gamification could have. These principles include spreadability vs. drillability, which refers to the balance between easy accessibility and deep engagement; continuity vs. multiplicity, which explores the relationship between a coherent narrative and diverse perspectives; immersion vs. extractability, which examines the level of audience participation and interactivity offered; worldbuilding, which involves the creation of immersive and expansive fictional worlds; seriality, which emphasizes the ongoing and episodic nature of storytelling; subjectivity, which acknowledges personal interpretation and individual involvement; and performance, which highlights the role of audience engagement and participation in the narrative experience.

Empirical studies, such as the one conducted by Ruiz-Bañulls et al. [22], have begun to indicate that the incorporation of gamified experiences in primary school classes, uniquely intertwined with the benefits provided by interdisciplinary work and transmedia narratives, significantly enhances students' training process and motivation, while also contributing to a better acquisition of compulsory curricular contents and improved academic performance. The interaction between gamification, interdisciplinary work, and transmedia narratives offers unique opportunities to enrich the learning experience and foster student participation.

Thus, the literature shows that the use of gamification in the educational environment can have important benefits, as it promotes logical thinking and improves students' motivation [23–26]. Students, who are familiar with the virtual environment and video games, recognize this pedagogical approach as playful and engaging. Although the literature points out that the success of this technique is not dependent on a specific profile, this has been demonstrated by studies carried out in different educational stages, from early childhood education to higher education [27,28] including primary education [29–31] and even secondary education [32], which show the effectiveness of this technique. Some studies, such as the one conducted by Guerrero-Puerta and Guerrero [8], also indicate that gamification can be especially effective with students in vulnerable situations, with the possibility of school failure.

The results of bibliometric analyses suggest that, although gamification is a widely studied technique in all educational stages, the use of gamification in the teaching of trainee teachers still has scarce research, although it is growing exponentially, so it can be considered a developing line of research due to the significant increase that has occurred in the subject in recent years [7,33].

Focusing on our object of study, through a recent systematic review [33], it has become possible to know the impact of gamified practices in the field of teacher training, both in its initial and permanent stage, evaluating the implementation strategies of elements of this pedagogical approach in the teaching–learning processes. This review has concluded that the use of gamification in the training of future teachers is a topic of great interest in the field of education, as the literature reveals that this pedagogical approach has become a

useful tool for improving the motivation, commitment, and participation of students in the teaching–learning processes.

The studies that explore gamification in teacher training have made it possible to delve into the main instructional design models for gamification systems. This has made it possible to establish a relationship with the elements implemented in the proposed practices, although in many cases the model involved in the design of the gamified practice has not been explicitly established [33]. In this sense, Navarro-Mateos et al. [34] points out that there is a general lack of knowledge about the process of gamification systems or specific models of instructional design on the part of teacher trainers, which causes the introduction of gamification elements without a specific criterion or without a configuration that has a specific purpose.

Despite the growing interest in gamification in academia, there is a lack of research focused on teachers' attitudes towards gamification and its actual implementation in teaching training. Marti-Parreño et al. [35] tried to address this gap by exploring teachers' attitudes towards gamification and their current utilization of gamification in higher education institutions. The findings indicate that only a small percentage (11.30%) of teachers regularly incorporate gamification in their courses, despite their overall positive attitude towards this approach. Moreover, the study reveals no significant differences in the use of gamification based on age, gender, or institution type (public or private). However, teachers serving in private universities exhibit a more positive attitude towards gamification compared to those in public universities. Interestingly, the results also suggest the existence of an "attitude-use gap," highlighting the disparity between teachers' positive attitudes towards gamification and its limited implementation in practice.

In this way, gamification, which in general terms has been shown to be a very positive tool as it increases the participation and motivation of participating students, seems not to be implemented with the systematization that might be expected from faculties of education [33].

Thus, according to González-Fernández et al. [33], an adequate educational approach to gamification in teacher training requires a deep knowledge of the implications derived from the implementation of this pedagogical approach. They point out that it is necessary to evaluate the importance of instructional design models so that they allow for the appropriate development of gamified practices, avoiding improvisation and arbitrariness. Although they point out that this technique is suitable for implementation in the field of teacher training, both in its initial and permanent stage, as the studies they have analyzed for their review show that the use of gamification involves experiential learning that allows teachers to introduce, in their professional development, this pedagogical approach in a relevant way, based on their own experience.

It seems paradoxical that the institutions that are normally responsible for the training of future teachers are not adopting these pedagogical innovations as quickly as one would expect. A review of the Spanish literature on gamification in teacher education shows that the application of this technique in teacher training is limited, or at least the research on its application is limited. There is still little research on the use of these techniques in teacher training, with the exception of the use of gamification in the area of specific didactics, where the technique seems to be having a greater impact, at least according to published studies. It is also significant that there is no reflection on the impact that the implementation of these techniques has on students with a view to their future as teachers, and the research focuses mainly on the design of the experience, or on specific applications, which is especially important considering that the studies are carried out in faculties of education, where the training of future teachers is an important part of the work carried out in these institutions [7].

Although there is limited research specifically on pre-service teachers, studies have begun to investigate teachers' perceptions, which indicate that gamified assessment is highly valued by teachers. The main advantages highlighted include increased motivation and the ability of gamified assessments to facilitate continuous learning beyond the classroom

setting. However, it is noteworthy that teachers express a lack of knowledge regarding gamification and its possibilities, despite having some understanding of game-based learning. This highlights the need for increased training on this pedagogical approach. Furthermore, it is important to note that there is still limited research focused on teachers in this field, emphasizing the need for further investigation [36–38].

The Implications that can be deduced from this theoretical framework are that gamification is a useful pedagogical approach to improve the motivation, engagement, and participation of students in teaching–learning processes. However, the literature shows that the use of this pedagogical approach in the field of teacher training is still scarce, although it is growing exponentially. This means that there is a great opportunity for educational institutions to adopt this pedagogical innovation to improve the training of future teachers. Furthermore, it is important that teachers have a thorough understanding of the implications of implementing this pedagogical approach, so that the design of gamified practices has a specific purpose and is not arbitrary. Finally, it is important that more research is conducted on the impact of the implementation of these techniques on students with a view to their future as teachers, which directly alludes to the objective of our case report.

### 3. Methods

#### 3.1. Study Design

This qualitative case report was conducted to explore the influence of contact/experience with gamification on prospective teachers' perceptions of future use of gamification. Specifically, the following research questions were selected as an initial guide:

1.  How do prospective teachers perceive the benefits of gamification as a tool to improve student learning?
2.  What feelings and emotions do future teachers experience during gamification?
3.  What training do future teachers consider necessary to be able to implement gamification effectively?
4.  What do prospective teachers consider important to improve the effectiveness of gamification?

#### 3.2. Sample

The sample consisted of three participants, selected using the snowball sampling procedure described by Noy [39]. The selection of participants followed specific criteria, necessitating that participants meet the following requirements:

*   Have undergone formal training in gamification as part of their teacher training process.
*   Possess prior experience with gamification as a participant.
*   Be currently enrolled in a teacher training program.

All participants received a prior explanation of the interview process and signed the corresponding letter of informed consent. It was impossible to continue the study for this reason, but given that this sample produced some insight into the impact that gamification has on prospective teachers, it is presented as a case report, the results not being generalizable but illustrative for further research that may be conducted on this topic.

#### 3.3. Conducting a Case Report

The case report is a widely used research approach in the medical field to present and analyze specific clinical cases [40]. However, this approach can also be applied in the field of education, where it is used to examine particular teaching and learning situations, as evidenced by previous studies [41–43] that have already highlighted the value of this study modality. It can be particularly relevant for exemplifying situations that have been underexplored in the literature but represent a significant example for further research in the field.

Thus, in medicine, cases are used to exemplify a situation that has been rarely reported but can be didactic. If we transfer this to the context of education sciences, a case report

refers to the detailed description of a specific case in which valuable lessons are analyzed and extracted for educational practice. These cases can involve various areas of education, such as classroom teaching, curriculum planning, educational program design, and student assessment, among others. In this case, we are situated in a meta-analytic case, in the sense that it explores the training of future educators from a qualitative perspective to understand how being exposed to gamification impacts this training and how it mobilizes or creates attitudes towards educational innovation, specifically towards the use of this pedagogical tool in the future.

To a large extent, the application of these cases in education is a tradition, as in the healthcare field, and many cases are published or used for educational purposes. The importance of case reports in education lies in their ability to generate practical and transferable, although not generalizable, knowledge that can be applied in different educational or research contexts [40]. By thoroughly analyzing a case, educators can identify best practices, common challenges, and possible solutions that can be useful for advancement. To achieve this, it is important for the paper to focus on the case description and its characteristics, without falling into over-analysis or a tendency towards generalization [43].

Although case reports in education may not have the same hierarchy of evidence as in medicine, they remain a valuable research form that complements more quantitative and experimental approaches. Case reports allow for in-depth analysis of complex educational situations and provide a unique and contextualized perspective that can contribute to the development of effective educational practices. Therefore, in the absence of the possibility of conducting a large-scale study and considering the observation of results that may be valuable if further investigated, the study is presented as a case report.

### 3.4. Data Collection Process

This study adhered to the principles of grounded theory methodology in designing the interviews. Semi-structured interviews were employed, allowing for flexibility in exploring emerging themes related to pre-service teachers' perceptions of gamification in their teacher education. The interview questions were open-ended and exploratory, enabling the collection of detailed and rich information regarding participants' individual experiences with gamification and their outlook on its future utilization. The probing technique was utilized during the interviews to delve deeper into meanings and gather additional details on pertinent topics. Additionally, constant analysis, a characteristic of grounded theory [44], was applied, facilitating the adjustment and refinement of interview questions as the emerging theory progressed. By following this rigorous and systematic approach, in-depth information was gathered, contributing to the understanding of pre-service teachers' perceptions of gamification in their teacher education.

Specifically, the interviews were conducted in an attempt to collect information on three main blocks (see Table 1): (1) educational experience of pre-services teachers, (2) analysis of pre-services teachers' perceived experience of gamification during teacher training, and (3) perception of the possibilities of gamification as a tool to be used in their future teaching.

For this purpose, it was decided to organize the data collection process according to the following structure:

- ➔ Introduction: Explanation of the purpose and scope of the interview.
- ➔ Educational experience of pre-services teachers
- ➔ Pre-services teachers' perceived experience: Targeted questions to find out respondents' experiences of gamification during teacher education.
- ➔ Perception of gamification: Questions aimed at finding out the respondents' perception of gamification as a teaching tool.
- ➔ Closing: Questions to close the interview, used to check some aspects arising from the data collection process.

**Table 1.** Methodology description.

| Format | Each interview: 1 h 15 min to 1 h 45 min |
| --- | --- |
| Format | Semi-structured |
| Flexibility | Open-ended questions |
| **Themes included in the initial schema of semi-structured interviews** | |
| **Educational experience of pre-services teachers** | Understanding participants' past experiences as students, focusing on pedagogical approaches, teaching methods, and overall school experiences. |
| **Analysis of pre-services teachers' perceived experience of gamification during teacher training** | Exploring participants' perceptions of gamification as a pedagogical approach during their teacher training. Investigating attitudes, emotions, and thoughts related to gamification, including motivational aspects, emotional responses, and assessment methods. Recognizing individualized experiences and differences in their accounts. |
| **Perception of the possibilities of gamification as a tool to be used in their teaching future** | Examining participants' views on gamification's potential as a teaching tool in their future careers. Insights on the applicability of gamification in different contexts and its role in enhancing student engagement. Emphasis on the need for further training to effectively implement gamification in future classrooms. |

Source: own elaboration.

*3.5. Data Analysis*

The collected data were analyzed using qualitative content analysis, transcribed, and analyzed using NVivo data analysis software. Recurring patterns in participants' responses were sought in order to identify trends and general conclusions. To ensure the reliability of the results, additional data collection was carried out, and feedback was provided to the participants so that they could correct any aspects that had not been correctly interpreted.

The codes and categories resulting from the study are as follows:

Codes:

➜ Positive attitude of teachers;
➜ Using games to teach;
➜ Equal treatment of students;
➜ A sense of achievement in overcoming challenges;
➜ Use of a narrative thread to permeate content;
➜ Use of timings and rankings to promote competition among students;
➜ Feelings of frustration and stress when facing challenges;
➜ Difficulty in avoiding stress and competition;
➜ Using the elimination method and re-engaging learners in gamification;
➜ Role of the teacher as a judge to make them reflect;
➜ Using a narrative thread to teach;
➜ Feeling of enjoyment in overcoming challenges;
➜ Student performance-based assessment.

Categories:

➜ Positive experience;
➜ Negative experience;
➜ Evaluation of progress;
➜ Feelings and sensations;
➜ General perception of gamification as a teaching tool.

The relation between these codes and categories can be observed in Figure 1.

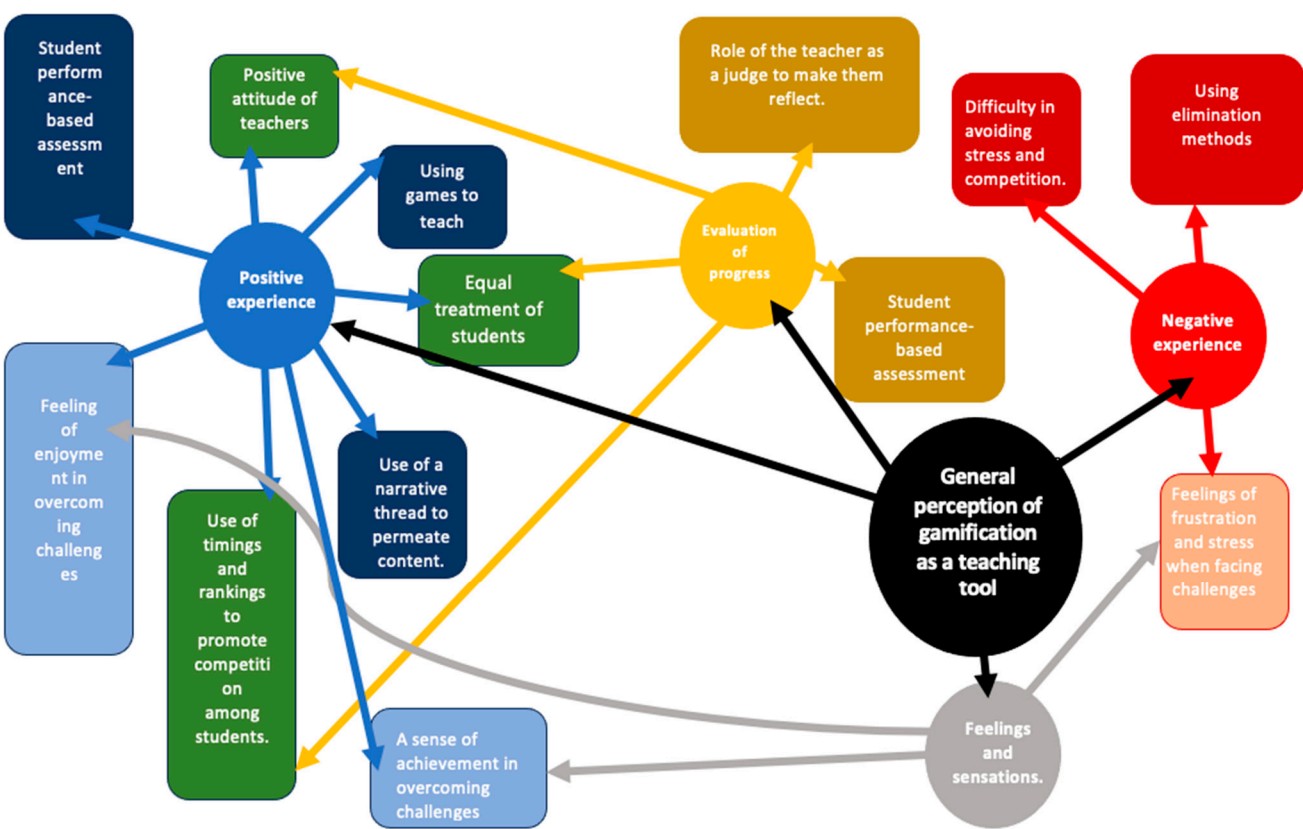

**Figure 1.** Relation between codes and categories. Source: own elaboration.

## 4. Results

This section presents the results obtained after conducting interviews with the three pre-service teachers, with the aim of examining their perception of gamification as a tool used in their teacher training. These have been organized according to the blocks described in the data collection process, and an initial block on the educational experience of the pre-service teachers has also been included in order to facilitate a better understanding of the sample.

### 4.1. Educational Experience of Students

The first theme emerging from the analysis is the educational experience of these preservice teacher (Concept-Indicator model could be seen in Figure 2).

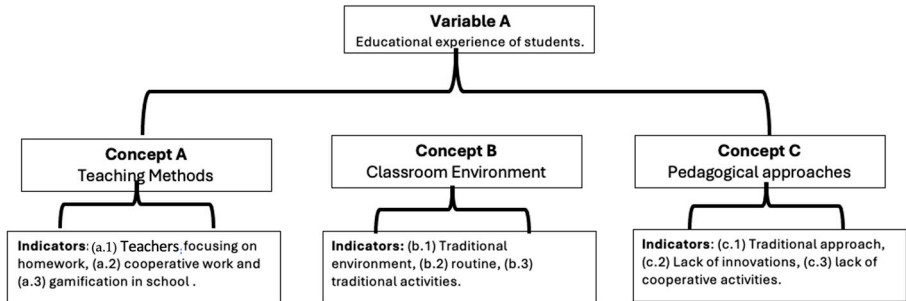

**Figure 2.** Variable A: concept-indicator model.

These three responses from pre-service teachers show a variety of experiences of school education.

Pre-service teacher 1 noted that he did not remember much from the first years of school, except that he had a bad time because the teachers focused more on giving

homework than on cooperative work or gamification. However, he remembered that in secondary school, he had some teachers who gave him more freedom, such as one teacher, Jaime, who allowed him to dedicate 45 min of class time to judo technique, although the subject was not related.

Pre-service teacher 2, on the other hand, pointed out that his school experience had been marked by constant changes, as he had to change schools at various stages of his education. In addition, he indicated that his school experience had been a traditional pedagogical approach, with content being explained and related exercises and activities being sent to him.

Pre-service teacher 3 also pointed out that, in his school experience, a traditional pedagogical approach had been predominant, without any innovative pedagogical approaches.

Thus, despite initial differences, these three pre-service teachers reflected similar school experiences, consisting of a traditional pedagogical approach, with content being explained, along with exercises and related activities. These school experiences did not include any innovative pedagogical approaches, such as gamification, cooperative work, or project-based learning (PBL).

### 4.2. Analysis of Students' Perceived Experience of Gamification during Teacher Training

The second theme emerging from the analysis is the Perceived Experience of Gamification during Teacher Training of these preservice teacher (Concept-Indicator model could be seen in Figure 3).

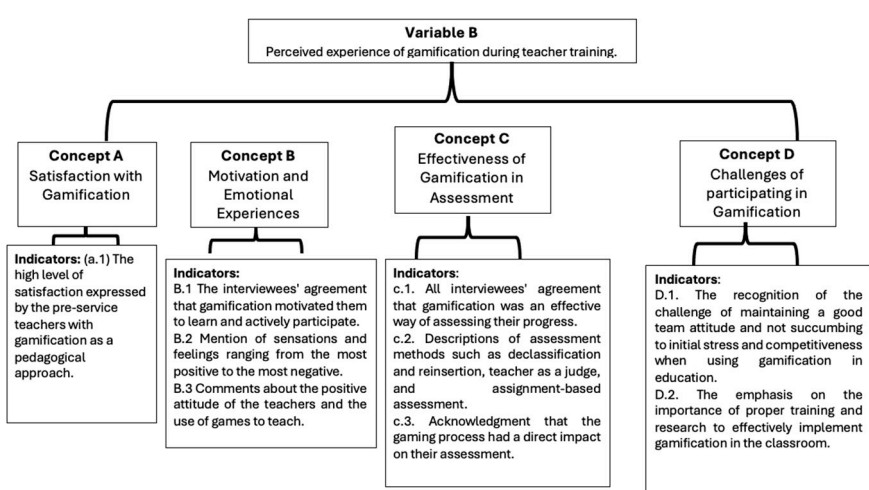

**Figure 3.** Variable B: concept-indicator model.

Interviews with the three pre-service teachers suggested a high level of satisfaction with gamification as a pedagogical approach. The interviewees agreed that this pedagogical approach motivated them to learn and participate, as well as providing sensations and feelings ranging from the most positive to the most negative. This can be seen in their comments about the positive attitude of the teachers, the use of games to teach, the equal treatment of the students, the feeling of overcoming challenges, and the use of a narrative thread to permeate the content. This can be clearly seen in the following interview excerpt:

> "I think it is quite positive, because in the subject I have been able to feel sensations and feelings from the most positive to the most negative. It is something that makes you get very involved, work very hard and that in many cases causes frustration and stress. The very positive part is that when you are able to overcome that stress and frustration, the feeling of overcoming it, the joy of having overcome it, makes what you have done and learnt very significant." (Pre-service teacher 2)

However, some differences between the accounts could also be noted. For example, pre-service teacher 1 focused on the use of timings and rankings to promote competition among the students, while pre-service teacher 2 highlighted the sense of frustration and stress that is generated when facing challenges. Pre-service teacher 3 focused on the use of a narrative thread to teach and the sense of enjoyment generated by overcoming challenges, and the difficulty of avoiding stress and competitiveness. These differences show that each interviewee experienced gamification differently, indicating that gamification offers a unique and individualized experience for each student.

In addition, all interviewees agreed that gamification was an effective way of assessing their progress. Pre-service teacher 1 mentioned that the teacher used the method of declassifying and reinserting students into the programme, depending on their performance, which ultimately influenced the final grade. Pre-service teacher 2 commented that the teacher was in the role of a judge to make them reflect on what they had done during the class session. Pre-service teacher 3 mentioned that the assessment was based on the students' performance on each assignment, which allowed them to get a final grade. But all agreed that the gaming process had a direct impact on their assessment, although they did not always show a thorough understanding of the criteria used for assessment.

> "What I mean is that, if you did better work, you got a better position in the house ranking, so to speak, what was called at that time the iron throne, in the Game of Thrones theme, and the iron throne was worth a 10 in the final grade. The rest of the houses, depending on their position within that ranking, were given a grade. So as far as I remember, there was no programme, no assessment tool, apart from this." (Pre-service teacher 3)

In conclusion, despite the differences, the interviewed have a common experience of gamification in education. They all recognize that gamification is a pedagogical approach that motivates them to learn and participate, as well as an effective way to evaluate their progress. Thus, they point out that gamification can be a useful tool for teaching, although they recognize that it is a challenge to maintain a good team attitude and not to get carried away by the initial stress and competitiveness.

*4.3. Perception of the Possibilities of Gamification as a Tool to Be Used in Their Teaching Future*

Lastly, the third theme emerging from the analysis is the Perception of the Possibilities of Gamification as a Tool to Be Used in the Future of these preservice teacher (Concept-Indicator model could be seen in Figure 4).

Through these three interviews, a general trend could be observed among prospective teachers of a positive perception of gamification as a tool to enhance and motivate learning. This trend could be seen in the responses of the interviewees, who highlighted the importance of gamification in making education more dynamic, interesting, and fun for students. It can also be noted that the interviewees shared the idea that, depending on the context, the use of gamification can be more effective than other pedagogical approaches. This can be clearly observed in the discourse of pre-service teacher 1, who explained his experience during the curricular practices as follows:

> "But it is true that once you get to the internship you see that everything changes and that there are many apathetic people, there are many people who are not motivated to use gamification and you find it a bit ridiculous, don't you? Because in the context I'm talking about, for example, there were three, four gypsy children and three, four children who were, you know, from lower neighbourhoods and I tried to make a gamification proposal for two or three days and the truth is that I didn't want to do it at all. So, if I had the right context, yes, I would like to do some cool things in terms of gamification, but especially with series and so on, you know? For example, la casa de papel or whatever, you know? Then, if the context is like this year's, I wouldn't possibly use gamification. I would try more to keep the children under control and so on." (Pre-service teacher 1)

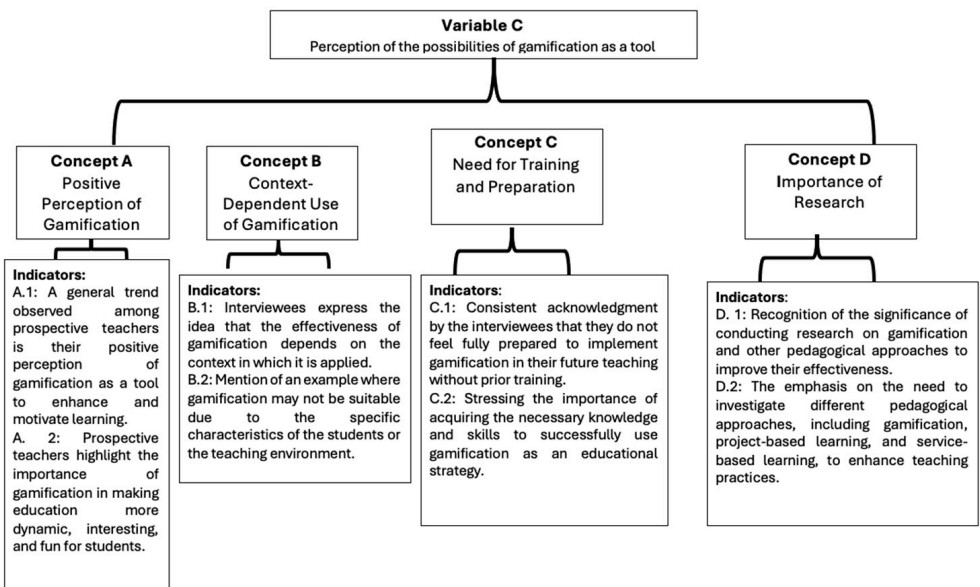

**Figure 4.** Variable C: concept-indicator model.

However, the idea also appeared consistently in the discourse of the interviewees that, although they experienced gamification during their teacher training, this took place as a practical experience, in which they themselves were the subjects who underwent the gamification process, so they did not feel prepared to carry it out in the future without prior training. Thus, they stressed the importance of researching both this and other pedagogical approaches to improve the effectiveness of the application of gamification in the classroom, or other innovative teaching techniques such as project-based learning or service-based learning.

In general, it can be concluded that prospective teachers have a positive perception of gamification as a tool to improve student learning, although they highlighted insufficient training. These interviewees also highlighted the importance of using gamification in the right context, as well as the importance of investigating other pedagogical approaches to improve the effectiveness of gamification.

Here is a summary table (Table 2) of the results obtained from the interviews with the three pre-service teachers regarding their perceptions of gamification as a tool in their teacher training:

**Table 2.** Summary of results.

| Section | Summary |
| --- | --- |
| Educational experience of students prior to teacher training | All three pre-service teachers had traditional pedagogical experiences during their school education. No innovative approaches like gamification were encountered. |
| Perceived experience of gamification | Pre-service teachers expressed high satisfaction with gamification, citing motivation, both positive and negative emotions, and effective assessment methods as key points. Individual experiences varied. |
| Perception of gamification in teaching | Pre-service teachers generally viewed gamification positively, recognizing its potential to make education dynamic and engaging. They emphasized the importance of context and additional training. |

Source: own elaboration.

## 5. Discussion

The results of the study provide valuable insights into the perceptions of pre-service teachers regarding gamification as a pedagogical approach in their teacher training. This

discussion will explore the implications and significance of these findings within the context of teacher education and gamification as a teaching tool.

One of the prominent findings from the study is the overwhelmingly positive perception of gamification among the pre-service teachers. They expressed satisfaction with how gamification motivated them to engage in learning activities and the range of emotions it elicited. This positive attitude aligns with existing research that highlights the motivational benefits of gamification in education [3,4,11].

The pre-service teachers acknowledged that gamification led to increased participation and engagement in the learning process. They appreciated the use of game elements to make the educational experience more dynamic and enjoyable. This finding is consistent with the idea that gamification can transform traditional teaching methods into interactive and engaging experiences, particularly for students who may have experienced conventional pedagogical approaches as monotonous [20].

Another noteworthy result is that gamification was seen as an effective method for assessing students' progress. The use of rankings and game-based assessment strategies was perceived positively by the pre-service teachers. However, there was also a recognition that a deeper understanding of assessment criteria is needed. This highlights the importance of aligning gamified assessment methods with clear learning objectives to ensure meaningful evaluation [3,4].

The study also revealed that the pre-service teachers recognized the importance of context when implementing gamification. They understood that the effectiveness of gamification could vary based on the specific classroom environment and student population. This insight underscores the need for educators to adapt gamification strategies to fit the unique needs and characteristics of their students [21].

A significant finding is that the pre-service teachers expressed a need for more comprehensive training in gamification. While they had positive experiences with gamification during their teacher training, they felt that additional theoretical grounding and practical guidance were necessary for them to implement gamification effectively in their future classrooms. This highlights a potential gap in teacher education programs, suggesting that more attention should be given to preparing future educators in the use of innovative pedagogical approaches like gamification [35].

The study also underscores the importance of continued research into the impact of gamification in teacher education. While the findings are based on a limited sample, they offer valuable insights and hypotheses for future investigations. As the field of gamification in education continues to grow, it is essential to examine how it can be integrated into teacher training programs more effectively [33].

## 6. Implications for Teacher Training

As we saw in the theoretical framework, gamification has become an increasingly used pedagogical approach in the classroom, providing a series of advantages for student learning. This has led to the need for specific training for teachers in charge of implementing this technique. In this sense, we present a reflection on the implications of gamification for teacher training derived from the case study that we present.

Firstly, it seems appropriate to point out that the results indicate that it might be necessary to develop specific teacher training for gamification, including research on the effectiveness of this pedagogical approach and the best methods for its application. This should not be limited only to gamifying a specific subject. The results of our analysis seem to indicate that this experience, without a theoretical accompaniment of the gamification process, benefits the students, fostering a positive attitude towards the technique, but without providing the necessary resources for future teachers to feel qualified to put it into practice in their future classrooms. Thus, future teachers should be trained in how to apply gamification in specific contexts, as well as the importance of researching other pedagogical or instructional approaches to improve the effectiveness of gamification, including thorough training on how to properly assess students' progress through gamification.

In addition, teachers need to be trained on how to motivate students through gamification, while avoiding negative effects such as stress and competitiveness. They also need to be trained on how to establish equal relationships between students, making everyone feel that they have the opportunity to participate and learn through gamification. It is important that this training helps future teachers to eliminate misconceptions about school contexts in which it is not possible to use this technique, especially when previous studies have indicated that gamification can be particularly effective with students in highly vulnerable situations.

## 7. Conclusions

In summary, the findings from the interviews with the three pre-service teachers provide valuable insights into their perceptions of gamification as a tool in their teacher training and its potential impact on their future teaching practices. Generally, all four initial research questions have been answered by this study:

- RQ1: The pre-service teachers overwhelmingly recognized the benefits of gamification in improving student learning. They highlighted its motivational aspects, positive emotional experiences, and its ability to engage students effectively.
- RQ2: The interviews revealed a range of emotions and feelings experienced by the pre-service teachers during gamification. While some encountered moments of frustration and stress, these were often overshadowed by the sense of accomplishment and joy that came from overcoming challenges.
- RQ3: The pre-service teachers acknowledged the need for additional training to implement gamification effectively in their future classrooms. They recognized that their exposure to gamification during their teacher training was a practical experience but felt that it required more comprehensive theoretical grounding.
- RQ4: Prospective teachers considered several factors important for enhancing the effectiveness of gamification. These included the context in which gamification is applied and the need for careful consideration of the student population. They also stressed the importance of researching and understanding various pedagogical approaches to complement gamification.

Overall, the responses from the pre-service teachers indicated a positive perception of gamification as a tool to enhance student learning. However, they highlighted the importance of further training and context-aware implementation. This research offers valuable insights into the potential of gamification in teacher training and its role in shaping the teaching practices of future educators.

In conclusion, we can point out that gamification is positioned in the discourse of the interviewees as a useful tool for teaching that offers a unique and individualized experience for each student. Pre-service teachers have a positive perception of gamification as a tool to enhance and motivate learning, although they also recognize that it is a challenge to maintain a good team attitude and not to get carried away by initial stress and competitiveness. This implies that future teachers need specific training to know how to motivate students through gamification, while avoiding negative effects such as stress and competitiveness, and to establish equal relationships between students. This training will also help to eliminate false beliefs about school contexts in which it is not feasible to apply this technique, especially when previous research has found that gamification can be particularly effective with students in highly vulnerable situations.

## 8. Limitations and Lines of Research for the Future

The main limitations of this study are the size of the sample and the fact that it is a case report. This means that the results obtained cannot be generalized to other contexts or situations. Therefore, it is recommended that studies with larger samples be carried out in order to obtain more general results. Furthermore, it is recommended that complementary studies with a quantitative approach be carried out in order to triangulate the results obtained in this study.

On the other hand, after this case report, new questions arise that point to new lines of research for the future. Thus, it is recommended to carry out studies that delve deeper into teachers' perceptions of gamification as a teaching tool, especially in relation to the use of this technique to improve the learning of students in vulnerable situations. Furthermore, it is recommended to investigate teachers' perceptions of gamification as a teaching tool, as well as to explore how this technique can be used to enhance learning in more disadvantaged educational settings, and how teachers' mindsets can be changed in this regard.

**Funding:** This research received no external funding.

**Institutional Review Board Statement:** The study was conducted in accordance with the Declaration of Helsinki, and the ethics procedures established by University of Granada and the Minister of Science of Spain (Code: LCTI, Ley 14/2011, 1 June 2011).

**Informed Consent Statement:** Informed consent was obtained from all subjects involved in the study.

**Data Availability Statement:** Data are unavailable due to privacy or ethical restrictions, but any information regarding data can be obtained if solicited from the author.

**Conflicts of Interest:** The authors declare no conflicts of interest.

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
