# Peer review of "Exploring if Gamification Experiences Make an Impact on Pre-Service Teachers’ Perceptions of Future Gamification Use: A Case Report"

_societies, doi:10.3390/soc14010011_

Round 1

Reviewer 1 Report

Comments and Suggestions for Authors

My main concern is that although the authors refer to gamification they do not define what is game and what is gamification (they give a brief version of the definition) for them and how they relate to each other, as well as what is gamification for the 3 pre service teachers.

In my opinion the entire analysis of the article refers to the role of game play in education and not to the role of gamification.

Additionally it does not seem that what is mentioned in the conclusions results from this particular research.

Some general comments:

*In the 'Legislative framework' chapter only the 1st paragraph corresponds to the title. The remaining paragraphs refer to the objectives of the work. I suggest to delete the chapter and integrate its content in the introduction chapter.

*It is not clear how authors connect "games based on transmedia narratives" and "interdisciplinary work' with the purpose of the manuscript.

*It is not clear how gamification is connected with assessment.

Comments on the Quality of English Language

*Exploring if gamification experiences make and  an impact on pre- 2 service teachers' perceptions of future gamification use: A case report.

*There are more than one full stops in a sentence in the text, full stops that are missing or that are placed in the wrong place.

*What "by foster student participation in primary education environments" means?

Author Response

Dear Reviewer, 

Thank you very much for all your recommendations, we have acknowledged all of them. This has tremendously helped to improve our manuscript. Below we attach a point by point response to your recommendations: 

My main concern is that although the authors refer to gamification they do not define what is game and what is gamification (they give a brief version of the definition) for them and how they relate to each other, as well as what is gamification for the 3 pre service teachers.In my opinion the entire analysis of the article refers to the role of game play in education and not to the role of gamification.It is not clear how gamification is connected with assessment.

We have added a footnote to clarify this cuestion, aditionaly we have included a discussion to better establish connection between theory and results. 

-Additionally it does not seem that what is mentioned in the conclusions results from this particular research.

New conclusion has been developed

*In the 'Legislative framework' chapter only the 1st paragraph corresponds to the title. The remaining paragraphs refer to the objectives of the work. I suggest to delete the chapter and integrate its content in the introduction chapter.

We have deleted this part

- Comments on the Quality of English Language: We have modified and proofread as recommended.   Thank you again,  Best regards,  Authors  

Reviewer 2 Report

Comments and Suggestions for Authors

This is an interesting study. The study is generally well-presented. Yet, there are some problems to be addressed:

1) What are the rationales for the study?

2) What is the itnerview guide like? What are the leading questions asked? in what langauge is the interview conducted? how long is each interview ... More info. is needed about the instrument.

3)It is not easy to find answers to the four questions in Results. Please do make it clearer

4) It is better to add Discussion to the manuscript.

Comments on the Quality of English Language

Generally good. Yet, there are some problems, e.g. "The a pre-service teacher 3 also ...". So please do carefully proofread the manuscript. 

Author Response

Dear Reviewer,

We sincerely appreciate your valuable feedback and have made the requested revisions to the manuscript. In this updated version, we have highlighted new information in green text and modifications in yellow. This visual distinction should make it easier for you to identify the changes that have been implemented.

Thank you for your helpful guidance, and we hope that these adjustments meet your expectations and improve the overall clarity of the manuscript.

If you have any further comments or suggestions, please do not hesitate to let us know.

Sincerely,

Author

Reviewer 3 Report

Comments and Suggestions for Authors  

This is an exceptionally intriguing topic. While there is a wealth of research exploring the effectiveness of gamification in teaching students, there has been limited investigation into the effectiveness of introducing gamification to pre-service teachers. The literature review is comprehensive, the language is fluent, the research purpose is well-defined, the research process is clear, and the overall readability is excellent.

1.  When introducing the definition, you mentioned that gamification is based on the use of game mechanics to motivate students. The original definition referred to "game elements." Additionally, "game mechanics" is a relatively complex concept; it would be beneficial to provide a brief explanation.

2.  I would appreciate a detailed description of the criteria used for selecting participants. What criteria led to the selection of the three individuals? It is advisable to provide an explanation.

3.  I suggest enhancing the readability of your "findings" section by including tables. I recommend referring to "Table 4" and "Table 5" in the paper titled "Factors contributing to teachers' acceptance intention of gamified learning tools in secondary schools: An exploratory study," particularly "Table 5."

4.  There are some minor formatting issues that require adjustment. On the seventh page, the two columns of bullet points under "codes" and "categories" are not aligned properly. Some have excessive indentation, while others do not.

Author Response

Response to the Reviewer's Comments:

Dear reviewer,

Thank you for your insightful feedback.

1. We have revised the definition of gamification to include "game elements" instead of "game mechanics" to align with the original definition. We have also added a brief explanation of "game elements" to enhance clarity for readers.

  1. We appreciate your suggestion regarding the criteria for participant selection. We have included a detailed description of the criteria used to select the three pre-service teachers in the "Sample" section, providing an explanation for the reader's better understanding.

  2. Thank you for your recommendation to improve the readability of the "findings" section. We have added a summary table at the end of the section, which should help readers grasp the key findings more easily. 

  3. We have corrected the formatting issues you pointed out on the seventh page, ensuring that the bullet points under "codes" and "categories" are aligned properly for consistency and clarity.

We sincerely appreciate your feedback, and these modifications have substantially improved the manuscript's quality. If you have any further suggestions or questions, please do not hesitate to let us know

Round 2

Reviewer 1 Report

Comments and Suggestions for Authors

The manusctipted was revised as suggested.

Author Response

Thank you very much, your review has been of great interest to the improvement of this paper. 

Best regards, 

Author